# A Composite Fabry-Perot Interferometric Sensor with the Dual-Cavity Structure for Simultaneous Measurement of High Temperature and Strain

**DOI:** 10.3390/s21154989

**Published:** 2021-07-22

**Authors:** Ping Xia, Yuegang Tan, Caixia Yang, Zude Zhou, Kang Yun

**Affiliations:** School of Mechanical and Electronic Engineering, Wuhan University of Technology, Wuhan 430070, China; ygtan@whut.edu.cn (Y.T.); yangcaixialy@126.com (C.Y.); zudezhou@whut.edu.cn (Z.Z.); yunkang@whut.edu.cn (K.Y.)

**Keywords:** Fabry-Perot, composite sensor, dual-cavity, high-temperature, strain, simultaneous measurement, photonic crystal fiber

## Abstract

In this paper, an optical fiber composite Fabry-Perot interferometric (CFPI) sensor capable of simultaneous measurement of high temperature and strain is presented. The CFPI sensor consists of a silica-cavity intrinsic Fabry-Perot interferometer (IFPI) cascading an air-cavity extrinsic Fabry-Perot interferometer (EFPI). The IFPI is constructed at the end of the transmission single-mode fiber (SMF) by splicing a short piece of photonic crystal fiber (PCF) to SMF and then the IFPI is inserted into a quartz capillary with a reflective surface to form a single-ended sliding EFPI. In such a configuration, the IFPI is only sensitive to temperature and the EFPI is sensitive to strain, which allows the achieving of temperature-compensated strain measurement. The experimental results show that the proposed sensor has good high-temperature resistance up to 1000 °C. Strain measurement under high temperatures is demonstrated for high-temperature suitability and stable strain response. Featuring intrinsic safety, compact structure and small size, the proposed CFPI sensor may find important applications in the high-temperature harsh environment.

## 1. Introduction

Strain is a vital parameter to characterize the mechanical and thermo-physical properties of the materials, thus accurate strain measurement has great importance within science and industry [1,2]. In many applications such as jet engines or power turbines, high-temperature manufacturing and nuclear power operation, extremely high temperatures and severe environments are encountered, which poses significant challenges to current high-temperature strain sensing technology [3,4]. At present, the most widely used high-temperature strain sensors are high-temperature strain gauges, which operates based on gauge resistance as a function of strain. However, they suffer from the inherent disadvantages of vulnerability to electromagnetic interference (EMI), mechanical hysteresis and creep [5] and drift in response due to oxidation [3], which diminish their reliability and accuracy in the above-mentioned environments. Therefore, it is a pressing demand to develop robust strain sensors capable of operating in environments with targeted high temperatures while maintaining a stable strain response.

As an emerging technology, optical fiber sensors offer an excellent solution to many of the above-mentioned challenges because of their unique advantages, which include immunity to EMI, chemical inertness, small size, lightweight, multiplex capability through one single fiber and allowing for flexible packaging. Over the past decade, variety of sensors based on optical fiber sensing technology have been developed and successfully applied in a wide range of fields such as civil structure, mechanical equipment, space aircrafts and robots, etc. [6,7]. Among them, numerous optical fiber strain sensors have been developed, which mainly include Fiber Bragg Grating (FBG) sensors [2,8] and Fabry-Perot (F-P) sensors [9,10,11]. FBG is wavelength encoded and it uses the linear response of Bragg wavelength shift to sense variations of temperature and strain [12]. When used for high-temperature strain measurement, total deformation including strain induced by thermal expansion as well as forces applied to the structure will be measured [13]. This temperature-strain cross-sensitivity may be a problem in practical applications, especially in high-temperature strain measurement which requires the discrimination of thermal strain and pure force-induced strain of the sensing point [5]. In order to overcome this issue, a number of techniques have been proposed before, such as the combination use of two FBGs [14,15], hybrid configuration of Bragg grating and long-period gratings [16], polarization-rocking filter [17]. However, the tensile strain limit of FBG is typically lower than 4000 με (~0.4%) [18], which limits the strain measurement range. Moreover, FBGs gets brittle and fragile [19] when exposed to high temperatures and thus would be easily damaged when improperly handled or subjected to harsh working conditions, leading to packaging difficulties for high-temperature strain measurement.

Compared with the other optical fiber sensing techniques, the Fabry-Perot interferometers (FPIs) are particularly attractive owning to high sensitivity, compact configuration, simple fabrication and good performance under high temperatures [20,21]. The FPIs have been realized in many different types [10,18,22,23,24] and can be divided as extrinsic Fabry-Perot interferometer (EFPI) and intrinsic Fabry-Perot interferometer (IFPI). In the configuration of the EFPI strain sensor, an F-P air cavity is introduced into the fiber to achieve strain measurement. The typical EFPI structure was formed by inserting two optical fibers with flat end-face into a glass capillary, with the fibers fixed to the capillary tube at two ends using epoxies or welding [25,26]. In such a structure, the strain measurement is realized based on the elongation of the capillary tube, thus it is not suitable for large strain measurement due to the vulnerability of the fixed points. Other structures of EFPI strain sensors have been realized by fabricating an air cavity inside the fiber with different shapes such as a micro-bubble [10,27], rectangular [28] or a spheroidal cavity [29]. In general, the EFPI strain sensors have high strain sensitivity and lower thermal sensitivity [30]. In the case of the IFPI strain sensor, fibers serve the F-P cavity and two in-fiber reflectors are formed to achieve strain measurement. Currently, most IFPIs are manufactured by fusion splicing different types of optical components, such as crystalline silicon at the end of SMF for temperature sensing [31] or photonic crystal fiber [32], multimode fiber [33] between two SMFs. The sandwiched structure IFPI can be used for high-temperature strain measurements, but it is sensitive to both strain and temperature simultaneously [10].

In this paper, an optical fiber Composite Fabry-Perot Interferometric (CFPI) sensor composed of a silica-cavity IFPI cascading a single-ended sliding air-cavity EFPI was presented with the capability of simultaneous temperature and strain measurement. The silica-cavity IFPI is fabricated at the end of SMF by fusion splicing a short piece of Photonic Crystal Fiber (PCF) to an SMF and the air-cavity EFPI is formed by inserting the IFPI to a quartz capillary with the reflective surface. In such a sensing structure, the EFPI detects total strain including thermal strain induced by the structure’s thermal expansion as well as mechanical strain induced by forces applied to the structure, while the IFPI is only sensitive to temperature without being affected by strain, which enables the proposed sensor to distinguish the thermal strain and force-induced strain accurately in the application of high-temperature strain measurement. Due to all-silica and single-ended sliding configuration, the proposed sensor shows high sensitivity and good linearity to strain with an excellent temperature resistance over the temperature ranging from 25 °C to 1000 °C. The temperature and strain testing have been conducted, which effectively verify the feasibility of the designed sensor for accurate measurement of temperature and strain in various applications under extreme environments.

## 2. Sensor Fabrication and Operating Principle

### 2.1. Sensor Design, Fabrication and Signal Demodulation

The configuration of the CFPI sensor is schematically shown in Figure 1a. The CFPI sensor head consists of two cascaded F-P interferometers: a silica-cavity IFPI with cavity length of *d*_1_ between the reflective face of R_1_ and R_2_, and an air-cavity EFPI with cavity length of *d*_2_ between reflective face of R_2_ and R_3_. A microscope image of the fabricated CFPI sensor is taken and presented in Figure 1b. It can be seen that the *d*_1_ is the length of the PCF, which has a pure silica solid core surrounded by a periodic array of microscopic air holes, and *d*_2_ is the length of the air cavity. As depicted in Figure 1c, when a broadband light *I_i_* incidents into the transmission SMF, the incident light will be partially reflected from R_1_, R_2_ and R_3_, as denoted as *I*_1_, *I*_2_ and *I*_3_, respectively. The reflected beams interfere with each other and generate there-beam interference. The intensity of the interference fringes *Ir* can be written as the following Equations (1)–(3).
(1)$$I_{r}=I_{1}+I_{2}+I_{3}+2\sqrt{I_{1}I_{2}}\cos(\varphi_{silica})+2\sqrt{I_{2}I_{3}}\cos(\varphi_{air})+2\sqrt{I_{1}I_{3}}\cos(\varphi_{air}{}_{+silica})$$
(2)$$\varphi_{silica}=\frac{4\pi n_{1}d_{1}}{\lambda }$$
(3)$$\varphi_{air}=\frac{4\pi n_{2}d_{2}}{\lambda }$$
where *φ_silica_*, *φ_air_*, *φ_air+silica_* = *φ_air_* + *φ_silica_*, are the round-trip propagation phase shifts corresponding to silica-cavity, air-cavity and the composite-cavity, respectively; *n*_1_(*n*_1_ = 1.444 for pure silica at 1550 nm) and *n*_2_(*n*_2_ = 1 for air) are the refractive indexes of PCF and air, respectively; *λ* is the light wavelength in a vacuum.

The fabrication process of the proposed sensor includes 4 steps in total, as depicted in Figure 2a–d in sequence. Firstly, a section of PCF (LMA-10, NKT Photonics, Birkerød, Denmark) was fusion spliced to the transmission SMF (G652D, YOFC, Wuhan, China). The fiber core of the SMF is fused silica, doped with germanium and the refractive index of the fiber core is ~1.45 at 1550 nm [34]. The LMA-10 is a single-mode photonic crystal fiber consists of a solid pure-silica core with the diameter of 10.1 μm surrounded by a periodic array of microscopic air holes. Secondly, the spliced PCF was cleaved to a short section of the desired length *d*_1_ with the help of a microscope. Meanwhile, the home-made sensor demodulator which can calculate the cavity length of the F-P interferometer was used for real-time monitoring of the sensor’s signal. Due to refractive index difference between the doped SMF core and the pure silica PCF core, Fresnel reflection occurs at the splice facet R_1_. Thus, the intrinsic F-P silica cavity with a cavity length of *d*_1_ was formed between the splice facet R_1_ and the cleaved end-face R_2_ of the PCF. Thirdly, a quartz capillary with an inner diameter of 128 μm for alignment and encapsulation was fusion fixed with a section of cleaved SMF at the right side. The cleaved end-face of the SMF serves a low-reflectivity end-face, marked as R_3_. The aforementioned arc fusions were carried out by a fusion splicer (S185 LDD, Furukawa, Tokyo, Japan). Finally, the encapsulation was achieved by inserting the silica-cavity IFPI to the capillary from the left side to form an EFPI of cavity length *d*_2_ with the reflective end-face R_3_, thus the CFPI sensor was formed. This step is realized through the high-precision fiber moving platform, which includes a three-dimensional micro displacement platform, a CCD camera and a computer. The IFPI is precisely moved by the micro displacement platform in order to penetrate the capillary and acquire the desired cavity length of *d*_2_. The CCD camera is used for the real-time observation of the optical fiber’s movement in the capillary.

A prototyped sensor has been fabricated and its interference spectrum is shown in Figure 3a. It can be seen that the interference spectrum of the CFPI sensor is a superposition of a coarse fringe pattern and a fine fringe pattern, which are resulted from the short silica-cavity IFPI and the long air-cavity EFPI, respectively. The spectral signal of the CFPI sensor was firstly processed and converted to the frequency domain by using the Fast Fourier Transform (FFT), the result after FFT is shown in Figure 3b. It can be seen that there are three peaks in the frequency domain spectrum which are denoted as peak 1, peak 2 and peak 3, corresponding to the frequency components of silica-cavity IFPI, air-cavity EFPI and composite-cavity (air plus silica) FPI, respectively. In order to obtain the independent interference spectra of the silica-cavity IFPI and air-cavity EFPI, two band-pass filters with the center frequency value of peak 1 and peak 2 were designed using Finite Impulse Response (FIR) filters to extract the frequency of interest. Then the two independent interference spectra which are the wavelength spectrum of the silica-cavity IFPI and the air-cavity EFPI were obtained using an Inverse Fast Fourier Transform (IFFT), as shown in Figure 4. After that, each interference spectrum was demodulated by employing White Light Interferometry (WLI) demodulation algorithm to compute the value of *d*_1_ and *d*_2_. For this sensor, *d*_1_ is 615 µm and *d_2_* is 3638 µm. 

### 2.2. Sensor Installation and Measurement Principle

After fabrication, the CFPI sensor needs to be assembled on the surface of the measured object to sense its deformation/strain and environmental temperature. Details of the installation of the sensor are shown schematically in Figure 5. The transmission fiber and the capillary bonded with reflective fiber are fixed to the measured object at fixed positions 1 and 2, respectively. The distance *L_0_* between them denotes the initial strain gauge. For the air-cavity EFPI, deformation of the measured part will lead to the transmission fiber’s axial slide along the capillary tube and change the cavity length *d*_2_, of the air-cavity EFPI accordingly. Thus, the corresponding strain variation can be calculated by demodulating the shift of the cavity length *d*_2_. However, for the silica-cavity IFPI, is free from strain.

Based on the mechanics of materials, the strain variation ∆ε measured by the EFPI can be expressed as follows:(4)$$\Delta \varepsilon =\frac{\Delta d_{2}}{L_{0}}$$
where ∆*d*_2_ is cavity length shift of the air-cavity EFPI, *L*_0_ is the initial gauge length. Equation (4) indicates that the cavity length-strain sensitivity of the EFPI sensor is determined by *L*_0_.

Temperature variation will induce the change of the phase shifts of the CFPI sensor and eventually results in the change of the interference pattern. For the silica-cavity IFPI, the refractive index of the guided mode in the silica cavity *n*_1_ and the cavity length *d*_1_ will change due to the variation of the environmental temperature ∆*T*, based on the thermo-optic effect and the thermal expansion effect, respectively. The variations of the refractive index ∆*n*_1_ and silica cavity length ∆*d*_1_ is given by:(5)$$\Delta n_{1}=n_{1}\cdot \xi \cdot \Delta T$$
(6)$$\Delta d_{1}=d_{1}\cdot \alpha_{f}\cdot \Delta T$$
where *α_f_* (*α_f_* = 0.5 × 10^−6^/°C) is the thermal expansion coefficient of silica, *ξ*(*ξ* = 8.3 × 10^−6^/°C) is the thermo-optic coefficient of silica [35], *n*_1_ (~1.444 for pure silica at 1550 nm) is the refractive index of the guided mode in the silica cavity. For such an IFPI, the corresponding functional relation of the change of Optical Path Difference (OPD = 2*n*_1_*d*_1_) with a temperature variation ∆*T* can be expressed as:(7)$$\Delta OPD=\Delta (2n_{1}d_{1})=2n_{1}\cdot \Delta d+2d_{1}\cdot \Delta n_{1}=2n_{1}d_{1}(\alpha_{f}+\xi )\cdot \Delta T$$

It can be seen from Equation (7) that there is a linear relationship between the silica-cavity IFPI’s OPD and ∆*T*, and the coefficient 2*n*_1_*d*_1_ (*α_f_ + ξ*) decides the OPD-temperature sensitivity of the IFPI. The temperature sensitivity of the IFPI with original cavity length of 651 μm is calculated to be 15.6 nm/°C. By demodulating the change of the optical path difference Δ*OPD*, the corresponding temperature variation ∆*T* can be obtained.

For the air-cavity EFPI, the length of the air-cavity EFPI will be changed due to the thermal expansion of the test part, which corresponds to the thermal strain of the test part. The thermal strain can be expressed as follows:(8)$$\Delta \varepsilon_{T}=\alpha_{P}\cdot \Delta T$$
where *α_p_* is the Coefficient of Thermal Expansion (CTE) of the measured part. It can be seen that the thermal strain depends on the material of the measured part and temperature variation.

Based on Equations (4) and (7), the total strain variation Δ*ε* and temperature variation Δ*T* can be simultaneously obtained by solving the following matrix equation:(9)$$\left[\begin{array}{c} \Delta \varepsilon \\ \Delta T \end{array}\right]=\left[\begin{array}{cc} k_{11} & 0\\ 0 & k_{22} \end{array}\right]\left[\begin{array}{c} \Delta d_{2}\\ \Delta OPD_{1} \end{array}\right]$$
where *k_11_* is the strain sensitivity of the air-cavity EFPI, which equals 1/*L_0_*, and *k_22_* is the temperature sensitivity of the silica-cavity IFPI, which equals 1/2*n*_1_*d*_1_ (*α_f_* + *ξ*).

In the practical application of high-temperature strain measurement, the strain variation measured by EFPI includes two parts: thermal strain *ε_T_* caused by the thermal expansion of the test part and mechanical strain *ε_M_* induced by the applied force. By subtracting the thermal strain from the total strain measured by EFPI, the pure mechanical strain could be obtained. Based on Equation (8) and matrix Equation (9), mechanical strain can be determined by solving the following matrix equation:(10)$$\Delta \varepsilon_{M}=\left[\begin{array}{cc} 1 & -\alpha_{P} \end{array}\right]\left[\begin{array}{c} \Delta \varepsilon \\ \Delta T \end{array}\right]$$

## 3. Experiments

### 3.1. High-Temperature Test

The CFPI sensor was evaluated for its temperature sensing characteristics from ambient to 1000 °C. The experimental setup for high-temperature test is shown in Figure 6. The experimental system consists of an amplified spontaneous emission (ASE) light source (83438A, Agilent, USA) with the wavelength range of 1500–1600 nm and output optical power of ~7 dBm (5 mW) at a wavelength of 1550 nm, a high-temperature furnace (OTF-1200X, KE JING, Hefei, China) and an OSA (AQ6370D, Yokogawa, Tokyo, Japan). The CFPI sensor with original silica cavity length (*d*_1_) of 615 μm and air cavity length (*d*_2_) of 3638 μm is chosen for temperature measurement. The tested sensor was mounted to the DZ125 superalloy metal plate and laid inside the glass tube in the central temperature zone of the high-temperature furnace to be heated. The furnace temperature was set to increase from room temperature (25 °C) to 1000 °C with an increment of 100 °C and kept constant for 30 min at each step to ensure a stable temperature, and then cooled down back to room temperature for 3 cycles. The broadband light from ASE light source injects the sensor and becomes reflected by it and goes back to OSA through the circulator. During the test, the spectra signal of the sensor was monitored and recorded by the OSA.

### 3.2. Strain Measurement and Calibration at Room Temperature

The strain measurement and calibration were performed using an equal-strength cantilever beam at room temperature. The experimental setup of the static strain measurement has been built, as shown in Figure 7. The CFPI sensor was pasted on the beam by use of epoxy adhesive, and a commercial resistance strain gauge was pasted near the CFPI sensor for a reference. During the static strain measurement calibration, weights from 0 g to 1000 g was loaded at the end of the cantilever beam at intervals of 100 g and then decreased back to 0 g. Such a procedure was repeated 3 times. During the experiments, signals of the CFPI sensor and the commercial strain gauge were acquired by a home-made demodulator (800 Hz) and a strain indicator (Donghua institute of electronics; DH-8302), respectively.

### 3.3. Strain Measurement under High Temperatures

A high-temperature strain experimental system based on a high-temperature tensile machine was built to test the high-temperature strain measurement performance of the CFPI sensor, as shown in Figure 8. The high-temperature strain experiments have been carried out at temperatures of 500 °C and 1000 °C, respectively, using the same experimental procedures. Before the test, the CFPI sensor was mounted on the DZ125 tensile specimen by metal bonding. The tensile specimen was fixed to the stretching device and put inside the high-temperature furnace. Firstly, the temperature was set to increase from ambient to the set value without load. When the temperature was stable at the set value, tensile loading test began, which included 6 steps: force loading (step 1)-constant load keeping (step 2)-force loading (step 3)-constant load keeping (step 4)-force loading (step 5)-constant load keeping (step 6). In every force loading step (steps 1, 3 and 5), the tensile strain of 550 με was gradually loaded to the specimen via a stretching device. After that, the load was kept constant for about 20 s (steps 2, 4 and 6). In the whole test, the reflection spectrum of the CFPI sensor was recorded by the home-made demodulator.

## 4. Experimental Results and Discussion

### 4.1. Results of Temperature Testing Experiments

The OPD-temperature response of the IFPI is shown in Figure 9a. The OPD variation from 25 °C to 1000 °C is 15.4 μm. The linear fitting of the average values shows the correlation coefficient (R^2^) of 0.998 and the OPD-temperature sensitivity of 16.12 nm/°C, which agrees well with the theoretical value of 15.6 nm/°C deduced from Equation (7). The error bars in the vertical axes at each temperature indicates the standard deviation for the measured OPD between test repetitions.

The thermal strain response of the EFPI under different temperatures was evaluated in Figure 9b,c. Figure 9b presents the thermal strain measured by the EFPI under different temperatures from 100–1000 °C. The maximum value of 16,215.33 με was measured under 1000 °C. It is noted that the curve of thermal strain-temperature is obtained by polynomial fitting, this is because the CTE of DZ125 [36] is not a constant value. For comparison, a Finite Element Method (FEM) simulation is performed to calculate the thermal strain response of the DZ125 test part under different temperatures from 100–1000 °C. A three-dimensional geometrical model of the DZ125 metal plate was built and imported to the static structural module in ANSYS. The material properties of DZ125 were used, including density, modulus of elasticity and CTE. The variations of the CTE due to temperature variations have been considered to properly predict the thermal strain response of the DZ125 metal plate to the applied temperatures. Figure 9c presents the comparison between the measured thermal strain value and simulated value, the slope of the fitting line indicates a good consistency between them.

### 4.2. Results of Strain Measurement and Calibration at Room Temperature

The static response of cavity length shift ∆*d*_2_ of the air-cavity EFPI under different loads, using the test setup described in Figure 7 is analyzed and presented in Figure 10. The maximum strain of 1088 με was measured. The average value of 3-round tests data was calculated and a corresponding linear fitting to the average value was analyzed. The cavity length-strain sensitivity of 0.0145 μm/με was obtained, with a correlation coefficient (R^2^) of 0.998. The vertical error bars at each strain value indicates the standard deviation of sensor’s cavity length shifts data between 3-round tests. The maximum deviation between the experimental cavity length shifts data is 0.252 μm, corresponding to a strain deviation of 17.38 με, which is 1.6% of its full range.

### 4.3. Results of Strain Measurement at High Temperatures

Figure 11a,b present real-time temperature and total strain data measured by the CFPI sensor under the temperature of 500 °C and 1000 °C, respectively. It can be seen that total strain (blue square line) including thermal strain and mechanical strain induced by the applied variable load according to steps 1–6 described in Section 3.3, was measured by the EFPI. At 500 °C, the maximum total strain of 8145 με was measured. The temperature value (orange star line) from 499.6 °C to 500.8 °C was measured by IFPI. Accordingly, thermal strain from 6422 με to 6436 με (red dot line) was obtained based on Equation (8). At 1000 °C, the maximum total strain of 17,480 με was measured. The temperature value (orange star line) from 999.2 °C to 1002.3 °C was measured by IFPI. Thermal strain from 15,657 με to 15,711 με (red dot line) was obtained based on Equation (8) correspondingly.

### 4.4. Results of Strain Measurement with Temperature Compensation

The temperature-compensated scheme employed in this paper was realized by subtracting the thermal strain from the total strain measured by EFPI, which enabled the discrimination of thermal strain and mechanical strain. Figure 12a,b shows the pure force-induced strain measurement value at 500 °C and 1000 °C, respectively, after temperature compensation by solving matrix Equation (10). It can be seen that the change in pure force-induced strain agrees well with the actual loading process described in Section 3.3.

At 500 °C, when actually applied strain is 550 με, the measured value is 532~543 με, with the max relative error of 3.3%. When actually applied strain is 1100 με, the measured value is 1109–1138 με, with the max relative error of 3.5%. When actually applied strain is 1650 με, the measured value is 1683–1709 με, with the max relative error of 3.6%. At 1000 °C, when actually applied strain is 550 με, the measured value is 553~574 με, with the max relative error of 4.4%. When actually applied strain is 1100 με, the measured value is 1119–1159 με, with the max relative error of 5.4%. When actually applied strain is 1650 με, the measured value is 1761–1775 με, with the max relative error of 7.6%. It can be seen that the deviation between the measured and actual loaded mechanical strain is below 10%. Such results validate the feasibility and effectiveness of the designed sensor for accurate measurement of mechanical strain under a high-temperature environment.

A few types of optical fiber FPI-based composite sensors for temperature-strain measurement under high temperatures have been demonstrated and compared in Table 1. The maximum working temperature and the maximum strain measurement value of the proposed sensor are the best among the other mentioned sensors.

## 5. Conclusions

In this paper, a novel optical fiber composite F-P interferometric (CFPI) sensor with the dual-cavity structure was developed to implement strain and temperature measurement under high temperatures. The CFPI sensor consists of a silica-cavity IFPI for temperature sensing cascading a single-ended sliding air-cavity EFPI adjacently for strain sensing. The temperature sensing principle of the CFPI sensor is based on the thermo-optic effect and the thermal expansion effect of the IFPI. The strain sensing principle of the CFPI sensor is based on the axial sliding movement of the EFPI with the deformation of the measured object. Due to all-silica and single-ended sliding configuration, the proposed CFPI sensor has good high-temperature resistance and a large strain measurement range. In the application of high-temperature strain measurement, the temperature was measured by the IFPI to acquire the thermal strain according to the thermal strain-temperature relation, meanwhile, the total strain including thermal strain and the mechanical strain was measured by the EFPI. By subtracting the thermal strain from the total strain, the discrimination of thermal strain and mechanical strain can be achieved. Such merits have important meaning for the application of high-temperature strain measurement since temperature-strain cross-sensitivity is inevitable, and accurate mechanical strain measurement is vital. To verify this capability, high-temperature strain experiments at 500 °C and 1000 °C with variable mechanical strain loaded have been conducted on the fabricated sensors, which demonstrate excellent performance in mechanical strain measurement with thermal strain interference eliminated.

Sensor installation is a key factor for contact measurement of strain. A reliable connection between the designed strain sensor and the measured object is demanded to ensure an accurate and reliable strain response of the sensor, especially for practical applications in severe working conditions. Metal welding has been implemented for installing the CFPI sensor to DZ125 test specimen in this paper for high-temperature application. The nickel-based GH3039 superalloy has been adopted as the welding material due to its stable high-temperature resistance and thermal match with the measured objects. Future work involves the optimization of installation procedures and parameters to improve the sensor’s accuracy and reliability. Additionally, the impact of different installation methods such as high-temperature adhesive and metal welding on the sensor’s performance will be investigated to provide the theoretical basis for the sensor’s error analysis.

## Figures and Tables

**Figure 1 sensors-21-04989-f001:**
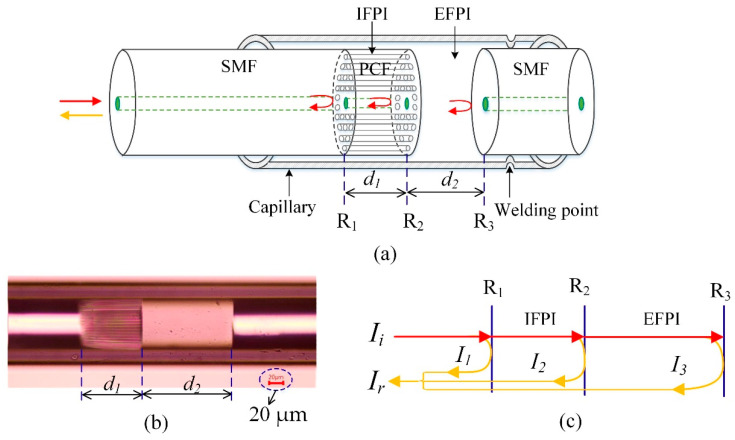
Schematic diagram of the CFPI sensor structure. (**a**) Concept diagram of the sensor structure; (**b**) microscope image of the fabricated sensor; (**c**) optical path illustration of the sensor.

**Figure 2 sensors-21-04989-f002:**
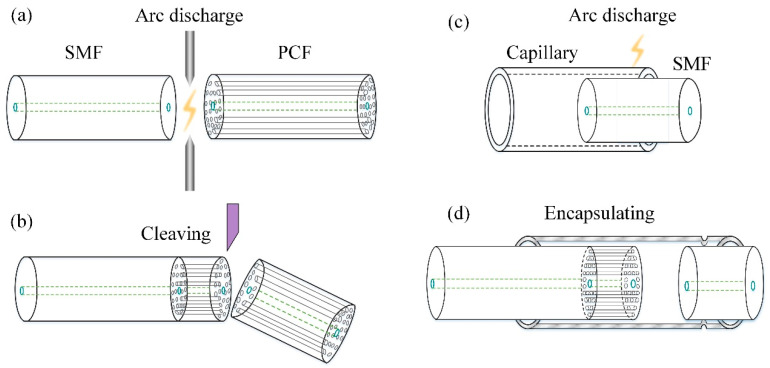
Schematic of the fabrication process of the CFPI sensor. (**a**) Arc fusion of the PCF to the transmission SMF; (**b**) Cleaving of the PCF to the desired cavity length; (**c**) Fusion fix of the reflective SMF to the capillary; (**d**) Alignment and encapsulation of the CFPI sensor.

**Figure 3 sensors-21-04989-f003:**
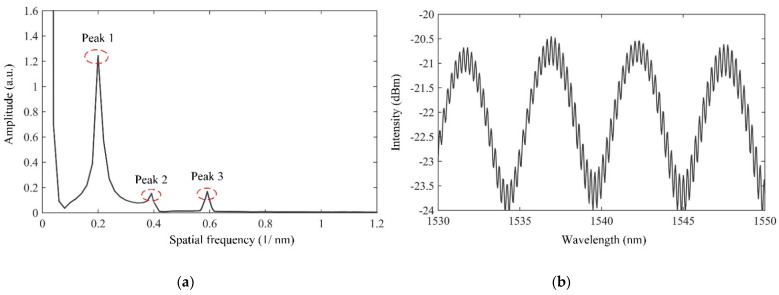
(**a**) Reflection spectrum of the fabricated CFPI sensor; (**b**) spatial frequency spectrum.

**Figure 4 sensors-21-04989-f004:**
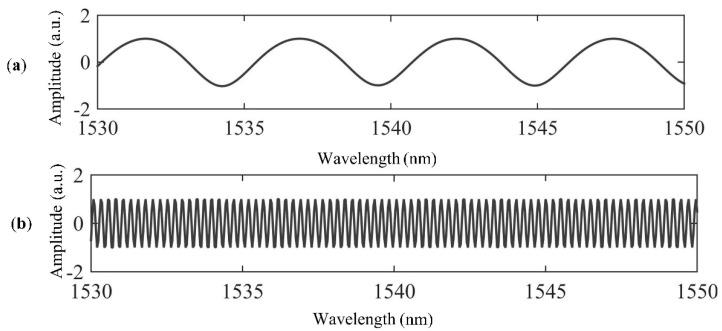
The wavelength spectra after FIR filtering, (**a**) the silica-cavity; (**b**) the air-cavity.

**Figure 5 sensors-21-04989-f005:**
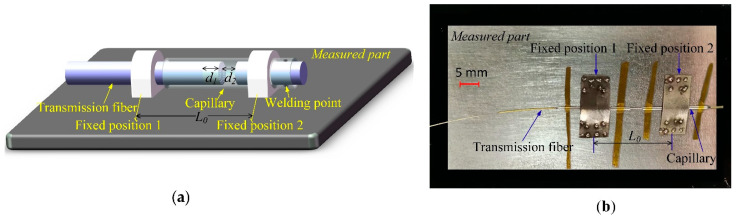
(**a**) Concept diagram of the sensor installation; (**b**) photograph of the sensor mounted on metal measured part by metal bonding.

**Figure 6 sensors-21-04989-f006:**
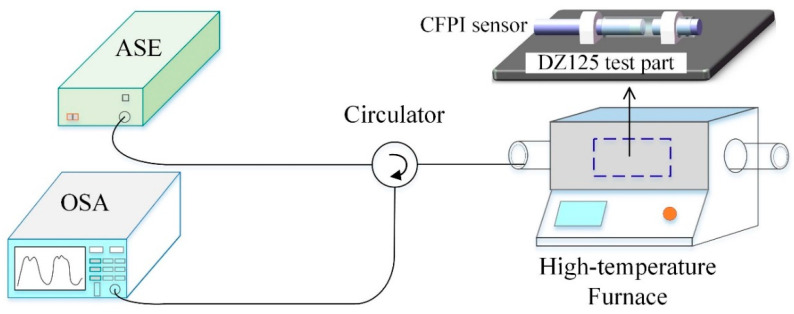
Schematic diagram for high-temperature test setup.

**Figure 7 sensors-21-04989-f007:**
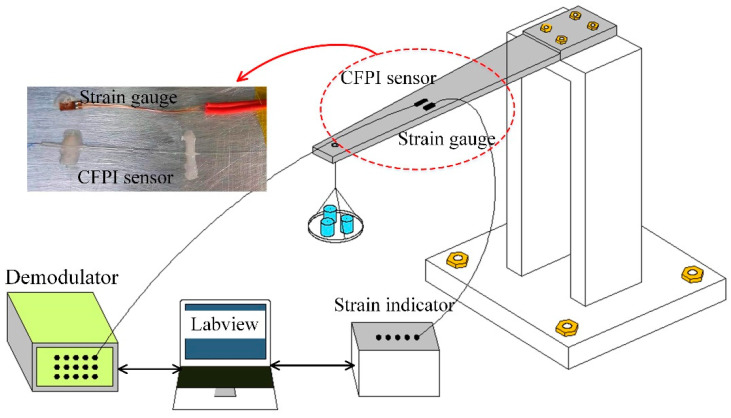
Schematic diagram of static strain measurement and calibration at room temperature.

**Figure 8 sensors-21-04989-f008:**
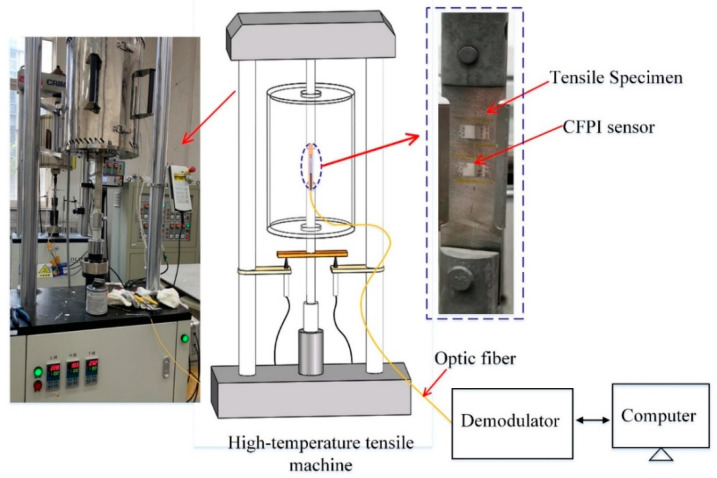
Schematic diagram of high-temperature strain measurement.

**Figure 9 sensors-21-04989-f009:**
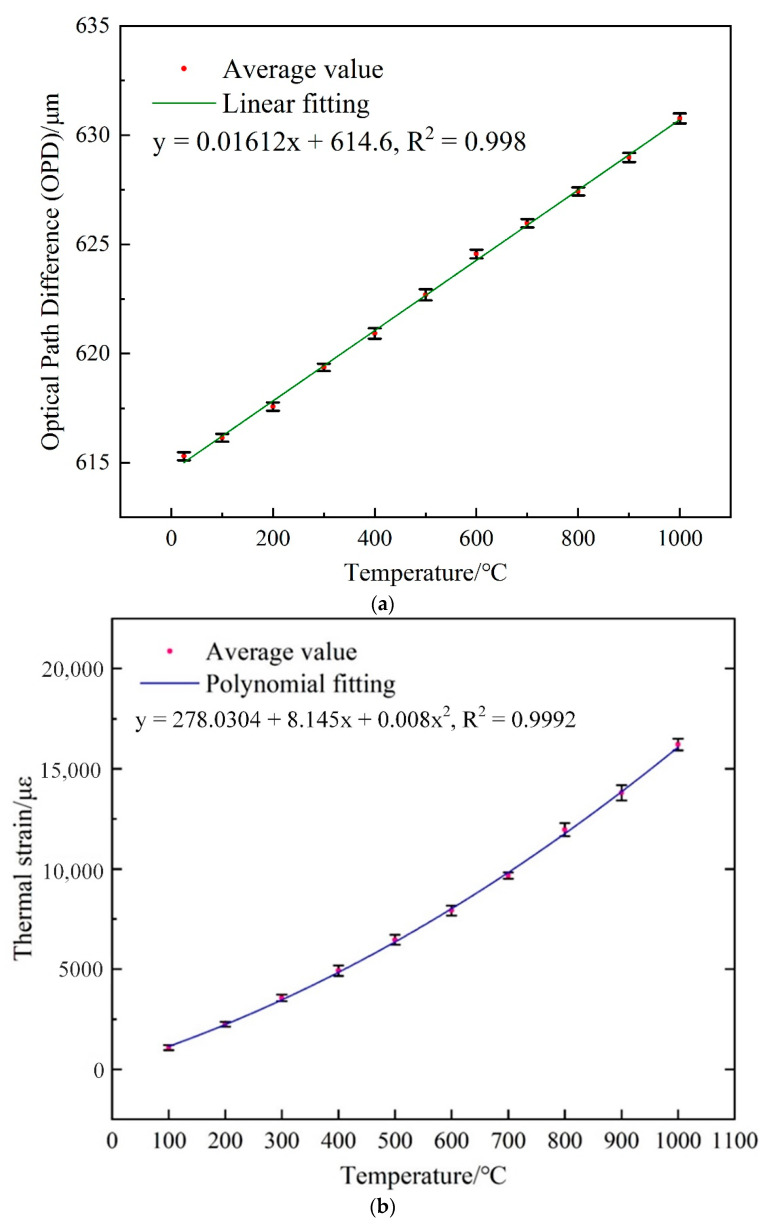
Temperature response of the CFPI sensor. (**a**) The relationship between the OPD of the silica-cavity IFPI and the applied temperature; (**b**) thermal strain measured by the air-cavity EFPI under different temperatures; (**c**) comparison between the measured thermal strain and predicted value by FEM simulation.

**Figure 10 sensors-21-04989-f010:**
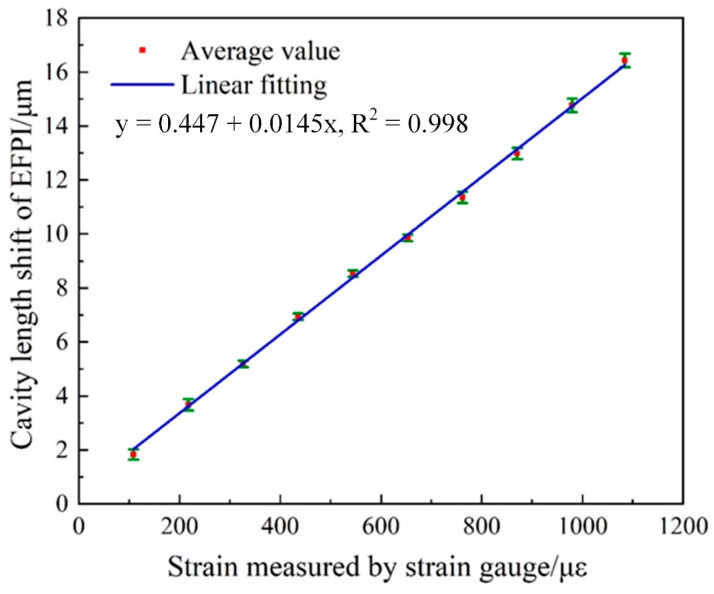
Cavity length shift (∆*d*_2_) of the air-cavity EFPI versus reference strain measured by the commercial strain gauge.

**Figure 11 sensors-21-04989-f011:**
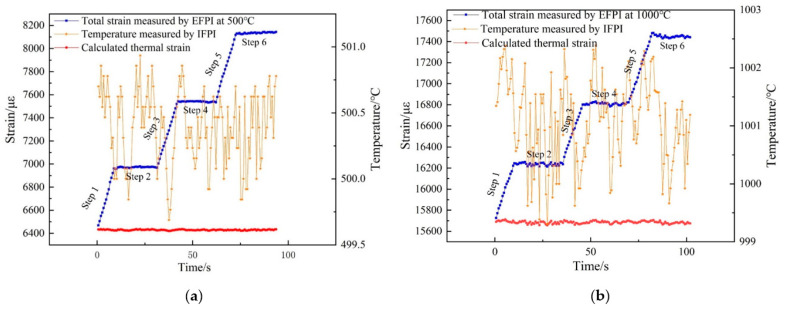
Real-time response of temperature and strain measured by CFPI sensor under different high temperatures with variable load, thermal strain calculated by Equation (8) as well, (**a**) under 500 °C; (**b**) under 1000 °C.

**Figure 12 sensors-21-04989-f012:**
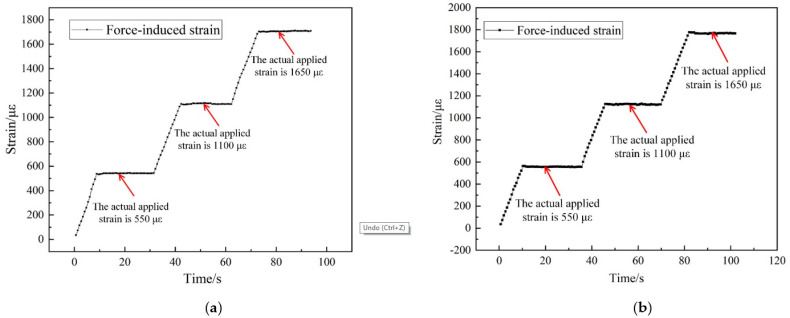
Results of high-temperature strain measurement with temperature compensation, (**a**) under 500 °C; (**b**) under 1000 °C.

**Table 1 sensors-21-04989-t001:** A few types of FPI-based composite sensors for temperature-strain measurement under high temperatures.

Research Group	Sensor Structure	Sensing Principle	Maximum Working Temperature/ Strain Measurement Range
Nan et al. [5]	EFPI-RFBG hybrid sensor	EFPI for strain measurement, RFBG for temperature measurement	800 °C/ 6500 με
Tian et al. [24]	FPI-RFBG hybrid sensor	FPI for strain measurement, RFBG for temperature measurement	1000 °C/ 450 με
This paper	EFPI-IFPI composite sensor	EFPI for strain measurement, IFPI for temperature measurement	1000 °C/ 17,480 με

## Data Availability

Not applicable.

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
