# Peer review of "A Composite Fabry-Perot Interferometric Sensor with the Dual-Cavity Structure for Simultaneous Measurement of High Temperature and Strain"

_sensors, 2021, doi:10.3390/s21154989_

Round 1

Reviewer 1 Report

For comments see attached file.

Reviewer 2 Report

The report presents a composite FPI sensor with dual-cavity structure for simultaneous measurement of high temperature and strain. I have some points to be addressed.

  1. Introduction needs to consider some literature with impact using FPI sensors for general sensing. Please read and consider: a) Analysis of viscoelastic properties influence on strain and temperature responses of Fabry-Perot cavities based on UV-curable resins, Optics & Laser Technology 120, 105743, 2019; b) Cascaded Fabry-Perot interferometer-regenerated fiber Bragg grating structure for temperature-strain measurement under extreme temperature conditions," Opt. Express 28, 2020; c) Fabry–Pérot curvature sensor with cavities based on UV-curable resins: Design, analysis, and data integration approach, IEEE Sensors Journal 19 (21), 9798-9805, 2019.
  2. How about the reproducibility? I see that repeatability tests were conducted and well addressed but how many similar probes were tested? Please add this info in the document because reproducibility between similar probes namely in cavities production is quite important and complex. This point is critical.
  3. How about the influence of humidity in the results? Please comment.
  4. How about the influence of L0, d1, d2 for the experiment and performance?
  5. How do authors reach to the optimal parameters for the cavity?

Round 2

Reviewer 1 Report

for comments see attached file

Reviewer 2 Report

The paper has been improved and can be published as it is.

Author Response

Thank you for your approval of our manuscript and we appreciate your suggestions to make the manuscript complete.

Round 3

Reviewer 1 Report

for comments see attached file.

Author Response

Comments to the Author

The paper reports on the design, fabrication, and properties of strain sensor for the high temperature applications. This is the third review of the paper and I have one more comment:

Q1: In my previous review, I asked to extend the paper about information for the applied photonic crystal fiber PCF and doped core of the SMF. The authors extended the paper about technical information for applied fibers, but I would like to ask you why was chosen fiber PCF with 10 cores? It is possible to use PCF with 8 or 12 cores?

Response:

The PCF of solid core with the diameter of ~10.1 μm was chosen because it’s single-mode PCF with the solid core. We use the PCF with the solid core to construct the IFPI because Fresnel reflection will occur at the splicing face when the PCF is fusion spliced to the SMF, due to the refractive index difference between the fiber core of the SMF and the PCF. Thus, theoretically, the PCF of solid pure-silica core with different diameters will be appropriate to construct the IFPI by splicing a short piece of PCF to the SMF. But it’s recommended to use the PCF with the diameter of 8~10 μm since it is single-mode fiber, and it has similar mode field diameter (MFD) with SMF we used, resulting in relatively small mode field mismatch loss and great increment of the reflectivity.

Final statement
The paper was significantly improved and after finale minor revision will be suitable for publication.

Response:

Thank you for your approval of our manuscript and we appreciate your valuable and constructive suggestions to make the manuscript complete.

Minor suggestions

Response:

We are sorry for the grammatical errors caused by our carelessness. In the revised manuscript, we have made significant efforts to remove the mistakes and errors and improved writing. All the errors you picked and recommendations you proposed are greatly helpful for us to polish our manuscript. We appreciate your elaborate efforts in reviewing. Thank you very much!